# A Machine Learning Model to Estimate Toxicokinetic Half-Lives of Per- and Polyfluoro-Alkyl Substances (PFAS) in Multiple Species

**DOI:** 10.3390/toxics11020098

**Published:** 2023-01-20

**Authors:** Daniel E. Dawson, Christopher Lau, Prachi Pradeep, Risa R. Sayre, Richard S. Judson, Rogelio Tornero-Velez, John F. Wambaugh

**Affiliations:** 1U.S. Environmental Protection Agency, Office of Research and Development, Center for Computational Toxicology and Exposure, 109 T.W. Alexander Drive, Research Triangle Park, NC 27711, USA; 2U.S. Environmental Protection Agency, Office of Research and Development, Center for Public Health and Environmental Assessment, 109 T.W. Alexander Drive, Research Triangle Park, NC 277011, USA; 3Oak Ridge Institutes for Science and Education, Oak Ridge, TN 37830, USA

**Keywords:** perfluoro-alkyl substances, PFAS, half-life, machine learning model, toxicokinetics

## Abstract

Per- and polyfluoroalkyl substances (PFAS) are a diverse group of man-made chemicals that are commonly found in body tissues. The toxicokinetics of most PFAS are currently uncharacterized, but long half-lives (*t*_½_) have been observed in some cases. Knowledge of chemical-specific *t*_½_ is necessary for exposure reconstruction and extrapolation from toxicological studies. We used an ensemble machine learning method, random forest, to model the existing in vivo measured *t*_½_ across four species (human, monkey, rat, mouse) and eleven PFAS. Mechanistically motivated descriptors were examined, including two types of surrogates for renal transporters: (1) physiological descriptors, including kidney geometry, for renal transporter expression and (2) structural similarity of defluorinated PFAS to endogenous chemicals for transporter affinity. We developed a classification model for *t*_½_ (Bin 1: <12 h; Bin 2: <1 week; Bin 3: <2 months; Bin 4: >2 months). The model had an accuracy of 86.1% in contrast to 32.2% for a y-randomized null model. A total of 3890 compounds were within domain of the model, and *t*_½_ was predicted using the bin medians: 4.9 h, 2.2 days, 33 days, and 3.3 years. For human *t*_½_, 56% of PFAS were classified in Bin 4, 7% were classified in Bin 3, and 37% were classified in Bin 2. This model synthesizes the limited available data to allow tentative extrapolation and prioritization.

## 1. Introduction

Per- and polyfluoro-alkyl substances (PFAS) are a large and diverse class of organic chemicals in which all (per-) or some (poly-) carbon–hydrogen bonds have been replaced with carbon–fluorine bonds [1]. Since carbon–fluorine bonds are stronger, they help make PFAS resistant to metabolism and degradation [2]. PFAS have both hydrophobic and lipophobic properties, from which they derive both water- and stain-repellant properties, thereby providing some of their utility to industry and consumers [3]. The majority of PFAS have either a straight- or branched-chain alkane backbone, with one or more functional groups bonded to the terminal ends of the backbone [2,4]. Examples of commonly studied straight-chained PFAS include carboxylic acids (such as perfluorooctanoic acid/PFOA) and sulfonic acids (such as perfluorooctane sulfonic acid/PFOS). A branched PFAS of note is GenX (perfluoro-2-methyl-3-oxahexanoic acid) [5,6]. Even the relatively well-studied PFOA and PFOS have lesser studied branched isomers [7,8].

PFAS are commonly found in human tissues [1]. Chemical properties of PFAS, such as the propensity to bind to protein, contribute to significant partitioning in the liver, the kidney, and the blood [9,10]. PFAS are of significant public health concern, as exposure has been associated with a growing list of pathologies in humans. Pathologies include endocrine system disorders, immunological disorders, fatty liver disease, cancers of the kidneys and testicles, and lower birth weight [11].

Due to the ubiquity of PFAS in body tissues, there is growing interest in characterizing the disposition of these chemicals within the body (that is, their toxicokinetics/TK) [12,13]. TK half-life (*t*_½_) is the amount of time needed for 50% of the chemical to be eliminated from the body. *t*_½_ is used to extrapolate from toxicological effects observed in animal species [14] and to understand human exposure [15,16,17]. Some PFAS (for example, PFOS) have been noted as having long half-lives (several years in humans). Widespread PFAS exposure from the environment and long half-lives result in the potential for bioaccumulation, as rates of uptake may exceed rates of excretion [18].

For typical organic chemicals, mathematical models exist for predicting properties related to human *t*_½_ from chemical structure [19,20,21,22]. However, these approaches are expected to fail for some PFAS [23] due to the peculiarities of fluorous chemistry [24] and potential biological interactions [25,26,27]. The estimation of PFAS *t*_½_ thus relies on either observational studies or extrapolation from animal species [11,28,29,30]. Typical extrapolation methods for TK parameters of PFAS are unreliable between species [14] and chemicals [27]. Efforts at extrapolating the measured PFAS *t*_½_ across species are complicated by unusual and unpredictable variability [26]. The *t*_½_ of perfluorohexanoic acid (PFHxA), for example, appears to scale allometrically (proportional to species weight) across mice, rats, monkeys, and humans [31]. In contrast, the *t*_½_ of the PFOA ranges from a few hours in female rats, days in male rats, 30–130 days in mice and monkeys, respectively [32,33,34], to 2–4 years in humans [35,36,37,38,39]. This large variation for PFOA occurs despite its structural similarity to PFHxA.

Under current chemical risk assessment paradigms, animals such as rats, mice, and monkeys serve as models to obtain toxicological information for other species where experiments may not be conducted; that is, humans and endangered wildlife. As toxicity testing evolves to include new approach methodologies [40], this may be less true. However, it is well known from physiologically based toxicokinetic modeling that understanding what phenomena can and cannot be extrapolated between species will inform human chemical risk assessment [41,42,43,44]. Thus, a key goal for PFAS is understanding differences in elimination kinetics between species [27].

Lau et al. [11,28,29,30] have reviewed the literature on in vivo measured interspecies PFAS *t*_½_ in 2007, 2012, 2015 and, most recently, in 2021. They have curated PFAS *t*_½_ data for multiple species across eleven PFAS. Most of the measured data are for rodents. While some PFAS rapidly transform to one of these eleven PFAS in vivo, [45] there are many thousands more for which there are no data available [12]. This is, in part, because in vivo experiments are resource intensive [46,47]. Additionally, higher throughput toxicokinetic methods perform poorly for some PFAS due to a lack of data characterizing transporters [23]. For linear PFAS only, *t*_½_ is observed to roughly increase with carbon chain length [36]. However, no systematic rules have been discerned for inter-species or inter-chemical extrapolation of PFAS *t*_½_ in general. Instead, each chemical and species require new in vivo studies [14,26,48]. Interaction with transporters and protein binding have both been suggested as relevant mechanisms that might be accessible in vitro [25,26,27,32], but these again require species- and chemical-specific measurements that are generally unavailable. Additionally, *t*_½_ varies with sex for some PFAS and species, with males typically having longer *t*_½_ than females [1].

Given the failure of typical approaches for the inter-species or inter-chemical extrapolation of PFAS *t*_½__,_ and the importance of this parameter for understanding the impact of these chemicals in the environment, a new approach is needed. Machine learning (ML) is an opportunity to use the available data to develop predictions for new chemical–species combinations. ML-based models of TK parameters can integrate multiple descriptors into predictive models for chemical properties [20,21,49]. Ensemble ML-based methods, such as random forest, combine predictions from an assembly of models (for example, regression/classification trees) to improve the robustness of the predictions. Each model contributing to the ensemble is built from a subset of predictors and/or training data records. Such ensemble models have been shown to provide reasonably accurate predictions over a range of chemical properties when empirical data are unavailable [20,50]. ML has previously been applied to PFAS, including to identify efficient treatment and removal from water [51] and to prioritize groundwater testing [52]. These prior works also used a variety of different machine learning approaches, including neural networks, the method of random forests, and other classification algorithms [51,52]. ML-based models might organize existing PFAS *t*_½_ data, categorize unmeasured PFAS, and identify the most impactful data needs for additional measurement. Since machine learning draws inferences from a data “training set”, one key metric for evaluating performance is a comparison of the difference between an ML model built with the actual training set and a model built using a “y-randomized” training set [53]. In y-randomization, the outcome to be predicted (in this case, *t*_½_) has been randomly swapped among the data. Y-randomization provides a baseline of how well a model might perform by chance.

In this study, we use the random forest method to develop a ML classification model for PFAS *t*_½_. We first use Monte Carlo methods to supplement the Lau et al. [11,28,29,30] *t*_½_ data set using TK studies not previously included. Given a small training set of eleven PFAS across four species, we aimed only to broadly classify PFAS chemical/species *t*_½_ into four categories: less than 12 h, 12 h to 1 week, 1 week to 2 months, or greater than 2 months. A diverse array of 119 descriptors was considered by the ML as potential predictors. These descriptors were mechanistically motivated, including both chemical and physiological properties. In particular, the descriptor set included several potential surrogates for transporters. Feature elimination was used to ensure a parsimonious model. To assess coincidental associations between descriptors and predictions, the actual model was contrasted against models built using multiple training data randomization approaches. We applied the model to a large set (~6600) of PFAS, for which *t*_½_ data are unavailable. Given the broad ranges of half-lives predicted by the model, for humans the model effectively predicts whether a given PFAS is more likely to be persistent. Those chemicals identified to likely be biologically persistent may pose an elevated risk. Finally, we use the predicted *t*_½_ values and a simple TK model to predict whole body clearance and steady-state plasma concentrations in multiple species.

## 2. Materials and Methods

The major steps of the workflow for this study included training dataset assembly, predictor set assembly, model construction, and model application (Figure 1). Dataset assembly is described in brief below, and in detail in the Appendix A. All analyses were performed using the freely available R statistical software platform v4.1.3 [54]. We used the following open-source tools (“packages”) from the Comprehensive R Archive Network (https://cran.r-project.org/, accessed 20 September 2022): caret [55], classyfireR [56], corrplot [57], data.table [58], gdata [59], ggplot2 [60], httk [61], MLmetrics [62], OneR [63], openxlsx [64], purr [65], randomForest [66], readxl [67], scales [68], showtext [69], stringr [70], and tidyr [71]. All scripts and data are available at: https://github.com/USEPA/CompTox-PFASHalfLife (accessed 17 January 2023).

### 2.1. Dataset Assembly

#### 2.1.1. PFAS Half-Life Data (Dependent Variable)

We modeled in vivo serum *t*_½_ data for 11 PFAS using published data experimentally collected from 4 species. The literature base was assembled from the most recent curation of Lau et al. (2021) [11,28,29,30] and supplemented with studies not previously reviewed. We intend models developed with these data to be preliminary attempts to classify the range of *t*_½_ of PFAS. Of the 11 chemicals, 6 are straight-chain perfluoroalkyl carboxylic acids: perfluorobutanoic acid (PFBA, DTXSID4059916), perfluorohexanoic acid (PFHxA, DTXSID30318623031862), perfluoroheptanoic acid (PFHpA, DTXSID1037303), perfluorooctanoic acid (PFOA, DTXSID8031865), perfluorononanoic acid (PFNA, DTXSID8031863), and perfluorodecanoic acid (PFDA, DTXSID3031860); 3 chemicals are straight-chain perfluoroalkyl sulfonic acids: perfluorobutanesulfonic acid (PFBS, DTXSID5030030), perfluorohexanesulfonic acid (PFHxS, DTXSID7040150), perfluorooctanesulfonic acid (PFOS, DTXSID3031864). The 2 remaining chemicals, perfluoro-2-methyl-3-oxahexanoic acid (GenX, DTXSID70880215) and perfluoro (2-((6-chlorohexyl)oxy)ethanesulfonic acid (F-53B, DTXSID80892506), are branched perfluoroalkyl carboxylic acids and perfluoroalkyl sulfonic acids, respectively. See the Appendix A for structural representations of each of the compounds. Chemicals and species were selected to have a range of data to inform extrapolation: species included humans, cynomolgus monkey (*Macaca fascicularis*), mouse (*Mus musculus*), and rat (*Rattus rattus*). Data from both sexes of each species were also included, as available.

Lau et al. [11,28,29,30] provide point estimates and ranges synthesizing multiple sources into consensus estimates of chemical- and species-specific *t*_½_. New peer-reviewed measurements were heterogeneously reported, including both measured and calculated mean *t*_½_ values per species, sex, and chemical that were usually accompanied by measures of variance (standard deviation, standard error, or 95% confidence interval). A Monte Carlo approach generated random samples using standard errors (SE) as the bounds of reported values/ranges. See Appendix A for details.

Distributions were generated by randomly sampling N animals (N = the sample size used in each estimate) from within the SE bounds assigned to each measurement, storing these samples in a vector, and then repeating this process 100 times. Each contributing study was represented in the complete vector of sampled values, in proportion according to sample size. Lastly, we fit a distribution to all samples and used the mean of this distribution as the *t*_½_ value in our training set for the corresponding chemical/species/sex/dosing method. Distributions were fit using the R package fitdistrplus [72], and an appropriate distribution (between the normal, lognormal, gamma, and exponential) as chosen based on the lowest AIC score.

Data were aggregated across multiple sources into a final dataset with a single value of *t*_½_ per chemical, species, sex, and dosing methodology; a total of 91 datapoints (Table 1). Of these, 50 were distinct measures by species and sex. See the Appendix A for the compiled processed dataset used for ML model construction.

#### 2.1.2. Chemical and Species Descriptors (Independent Variables)

We assembled a set of 119 chemical and physiological descriptors as potential predictors of *t*_½_ in ML models. These descriptors characterized either the structure of the chemical agent or the physiology of the animal species; please see full details in the Appendix A. We use the term “predictor” for chemical descriptors that are identified as predictive by ML.

Physico-chemical descriptors (22 descriptors): Physico-chemical descriptors have been shown to characterize TK for organic chemicals present in pharmaceuticals, elsewhere in commerce, and the environment [20,21,91,92,93]. Here, 18 physico-chemical properties predicted by version 2.7 of the OPERA modeling platform [50] were used. We note that OPERA’s training sets were recently updated to include additional PFAS data on LogP, water-solubility, vapor pressure, and melting point (https://github.com/kmansouri/OPERA/releases/tag/v2.7-beta2, accessed on 1 October 2021). In addition, some PFAS have been designed to include an ether bond to potentially facilitate more rapid metabolism [94]. To account for this, a binary descriptor (the ToxPrint Chemotype [95] “COC_alphatic”) was included, denoting the inclusion of an ether bond along the carbon backbone. Finally, average molecular mass and two chain length descriptors were included.

Transport/re-uptake analogs (88 descriptors): Although some PFAS are metabolically stable, they may still be subject to active cellular transport by the body, particularly if they are mistaken for endogenous, non-fluorinated analogs. For example, the long half-life of PFOA in humans has been attributed to reabsorption in the kidney by transporters for the endogenous caprylic acid [26,96]. Unfortunately, PFAS-specific transporter affinities [25,97] and species-specific data on variation on transporter ontogeny [98] are often unavailable. As surrogates for species- and chemical-specific data on the expression of relevant transporters, we examined two types of potential predictors:

Physiological descriptors including kidney structural features as surrogates for renal transporter expression (21 descriptors): The kidney is suspected to be a primary site of PFAS elimination and active transport (secretion/reabsorption) [96,99]. While the species- and chemical-dependent affinities for the transporters driving section/reabsorption are not typically known [26], they are expressed along the surface of the proximal tubule, and so geometry provides one available descriptor that might be correlated with clearance [100], in this case by limiting the surface area available for the expression of transporters. To capture the potential of physical aspects of the kidney as a surrogate for the amount of active transport, a suite of 21 kidney structure descriptors (for example kidney weight, number of nephrons, glomerular surface area) was assembled from Oliver [101] which reported these properties for rat, rabbit, dog, human, cattle, elephant, whale, horse, and chicken. Regressions were made on log-transformed body weight and these regressions were used to make predictions for mouse and monkey based upon body weights reported by Davies and Morris [102] (see GitHub file “CurrentScripts/1_PFAS_Dataset_building.R” for additional information). Overall species body weight was also included as a potential predictor, but was found to be heavily correlated by feature elimination (below).

The similarity of “Defluorinated” PFAS to Endogenous ligands as surrogates for transporter affinity (67 descriptors): As an additional surrogate of the impact of active transport on PFAS, we considered the structural similarity of defluorinated PFAS and a set of 894 endogenous compounds [103] that might be transporter substrates. We presume that structural similarity might result in exogenous chemicals serving as ligands for transporters of endogenous chemicals [104]. Several PFAS have similar non-fluorinated endogenous analogs; for example, caproic acid (that is, Hexanoic acid, CASRN:142-62-1, DTXSID7021607) may be a substrate for human peptide transporter 1 (PEPT1), which facilitates renal reabsorption of peptides in the proximal tubules of the kidney [105,106]. Caproic acid is structurally equivalent to perfluorohexanoic acid (CASRN: 307-24-4, DTXSID:3031862), with hydrogen atoms instead of fluorine atoms along its carbon backbone. To incorporate this information into a predictor dataset, we calculated molecular descriptors (PubChem and Morgan fingerprints) for PFAS in which each fluorine was replaced with hydrogen. Then, we calculated Tanimoto [107] scores (that is, Jaccard similarity) between the defluorinated PFAS and the endogenous compounds for each fingerprint. The subset of endogenous compounds with the highest and lowest similarity for each PFAS was then selected as potential predictors. In this subset, similarity values were discretized (>0.9 being similar (1), otherwise dissimilar (0)) and used as values for each predictor. Among the 11 structures, there were 65 endogenous ligands with at least one non-zero descriptor plus the two maximum values across all ligands for PubChem and Morgan.

Protein Binding (4 descriptors): PFAS bind to specific proteins in the liver and to albumin in serum, which likely influences clearance rates (and therefore *t*_½_) [29]. To account for this, two experimentally available serum–albumin binding rate constants, K_a_ (M^−1^) [26], and two binding rate dissociation constants to the liver fatty acid binding protein (L-FABP) [108] were added for a subset of PFAS where measurements had been made.

Categorical Descriptors (2 descriptors): We considered sex (male, female) and dosing type as indicated in the literature source documentation (intravenous, oral, other (epidemiological, via metabolite extrapolation)).

#### 2.1.3. Descriptor Reduction

The total descriptor set (119) was reduced prior to modelling; see Appendix A for full details. First, we identified and eliminated low variance predictors—that is, those predictors that have nearly the same value for most chemicals—defined as predictors with standard deviation/mean < 0.05. Next, we eliminated highly (>0.9) correlated predictors using the “findCorrelation” function of the caret [55] package of R statistical analysis software. This resulted in 13 numeric descriptors plus the two categorical descriptors that were held out of the quantitative analysis. A summary of the 15 descriptors used is shown in Table 2. Prior to modeling, these were mean-centered and scaled by standard deviation.

### 2.2. Model Development

We used the R caret package [55] to iteratively call the randomForest package [66] to construct random forest [109] classification models of *t*_½_ using all 15 independent descriptors. The classification approach was selected due to the limited size (91 data points) and scope (11 chemicals) of the training set. All models described below were fit using 10-fold cross validation with 10 repetitions at each step. We evaluated 3, 4, and 5 bin models. Bins were initially split into approximately equal proportions using the OneR package [63]. Bins were slightly adjusted towards whole number time increments. The distribution of data points into the bins was similar, ranging from 22.0 to 29.7%.

To further reduce overfitting, we used recursive feature elimination (“rfe” from caret [55]) to find the model with the highest accuracy with the fewest of the 15 descriptors in Table 2. Starting with the full 15 descriptor set, a series of models were built using sets of progressively fewer descriptors, with the least “important” descriptor excluded from one series to the next. Predictor importance [109] was quantified as the percentage reduction of model accuracy resulting from permutation of that particular predictor.

### 2.3. Model Evaluation

Machine learning involves a set of data used to construct the model (a training set) and a second set of data used to evaluate the model (a test set). Our ability to evaluate models was limited, as insufficient data were available to formulate a test set. To partially evaluate the performance of the models, we employed y-randomization; in this case, y-randomization tests for false associations by randomly permuting the *t*_½_ half-life categories, while keeping the descriptors the same. We then refit the model using the same methodology as for the training set. For each y-randomization approach we considered, we built ten models using ten different y-randomized data sets. To evaluate how the distribution of variance of *t*_½_ values between species, between chemicals, and between chemicals and species influences model fitting, this process included *t*_½_ values y-randomized in three ways. First, *t*_½_ values were randomized across all species and chemicals. Next, *t*_½_ values were randomized between species of the same chemical. Third, *t*_½_ values were randomized between chemicals of the same species. Finally, we computed and compared model accuracies between the models constructed using the three types of y-randomized values and non-randomized *t*_½_ values.

The prediction of error of the random forest models was characterized using out-of-bag (OOB) error—each decision tree of the random forest is constructed with a randomized subset (in-bag) of the available data and the data withheld from that tree’s construction (OOB) are used as a test set to evaluate the performance of that tree. OOB error of the ensemble of trees (that is, the random forest) is the average OOB error across the ensemble. For a categorical (classification) model, a confusion matrix can be constructed in which each row of represents the instances of the correct class for the samples from a test set, and each column represents the predicted class for samples—a perfect predictor would only have values on the diagonal. For a random forest model constructed with R package randomForest, a confusion matrix is calculated using the OOB data only. Finally, for a categorical model, a “No Information Rate” is calculated as from the largest class percentage in the data set, representing the performance of a “model” in which all samples were predicted to be in the most commonly occurring class.

### 2.4. Model Application

#### 2.4.1. Prediction of Half-Lives for Novel Chemicals and Species

The *t*_½_ model was applied to the largest list of PFAS available from EPA’s CompTox Chemicals Dashboard (CCD) [110] (https://comptox.epa.gov/dashboard/chemical-lists/pfasmaster, accessed on 1 January 2023). PFASMASTER is “a consolidated list of PFAS substances … of current interest to researchers and regulators worldwide” that includes PFAS from multiple EPA lists, the OECD New Comprehensive Global Database, KEMI Swedish Chemicals Agency Report, and the NORMAN Suspect List Exchange, among others. This is a list of 8163 PFAS compounds (as of August 2020) with structural information that is listed on the USEPA CompTox Chemicals Dashboard [110]. Predictor values for these compounds were assembled in a similar way to the training set. When a predictor value was unavailable for a chemical, average values were imputed from available data, resulting in some predictors being largely imputed from a small subset of available chemicals (for example, serum–albumin-binding coefficients). In addition, the model was applied to a new species, the domestic dog (*Canus domesticus*), to demonstrate its applicability to a novel species based on changing the model’s kidney predictor values. The distribution of the predicted *t*_½_ of the chemicals was plotted for both models for each species.

The applicability domain (AD) of the model was characterized using the methodology of Roy et al. [111]. This method considers whether the distribution of the scaled descriptors of a novel chemical are captured within the distribution of the training chemical descriptors. Each chemical of the CCD PFAS list was described as either inside or outside the domain of the *t*_½_ model by species. In addition, several predictors were chemical properties estimated with OPERA models, and thus had their own ADs. Thus, each chemical by species was further delineated by whether it was included in the domain of both the *t*_½_ model and all underlying predictor models. We describe the intersection of “All Model ADs” as the “AM domain”. Lastly, we used the chemical classification tool ClassyFire [56] to help characterize the predicted chemicals relative to the chemicals in the training set.

#### 2.4.2. Prediction of Serum Concentration

Finally, we used *t*_½_ predictions within a simple 1-compartment model framework to predict steady-state concentrations within the body following exposure. This process included first using predicted *t*_½_ values to calculate elimination rate constants (*k_elim_*, Equation (1), units of h^−1^), which are then used to calculate whole body clearance rates (*CL_tot_*, Equation (2), units of L/kg body weight/day) and whole-body, steady-state concentrations (*C_ss_*, Equation (3), mg/L):(1)kelim=ln2t12
(2)$$CL_{tot}=V_{d}\times k_{elim\;}\times 24\;$$
(3)$$C_{ss}=\frac{D}{CL_{tot}}$$

In Equation (2), the volume of distribution (*V_d_*) can be defined as the volume needed to yield the concentration of a chemical observed in plasma [112]. To estimate *V_d_* across chemicals and multiple species, we investigated developing models using the same process as for *t*_½_; see Appendix A for further details. In Equation (2), the factor of “24” allows *CL_tot_* to be given in units of in L/kg body weight/day. In Equation (3), steady-state plasma concentration (*C_ss_*) is calculated by assuming a constant dose rate (D) of 1 mg/kg body weight/day, which may be then used for reverse dosimetry in vitro–in vivo extrapolation [61]. Using this approach, we predicted steady-state concentrations *C_ss_* for each species (including the inferred species, dog (*C. familiaris*)) for PFAS compounds for which QSAR-ready SMILES were available for descriptor calculations, and which fell into All ADs of the model.

## 3. Results and Discussion

### 3.1. Half-Life Model Optimization and Selection

Knowledge of chemical-specific *t*_½_ is necessary for exposure reconstruction [15,16,17] and extrapolation from toxicological studies [14]. For PFOA, we found the TK *t*_½_ scales only weakly across species with bodyweight (R^2^ = 0.39). This scaling was on average even less for the other chemicals in our data set (R^2^ = 0.26 overall). Instead, a total of 119 descriptors (including body weight) was considered for modeling *t*_½_. The number of descriptors was reduced prior to modelling; see Appendix A for full details. First, correlation was used as a guide to identify 15 independent descriptors; for example, both body and kidney weight were identified as highly correlated with other physiological features and eliminated. For the 15 descriptors, listed in Table 2, models were constructed iteratively using subsets of the 15 descriptors. This recursive feature elimination process did not further reduce the number of predictors. That is, a model built using all 15 predictors was identified as optimal. We used ML to organize the available in vivo PFAS TK *t*_½_ data into to three, four, or five bins using the predictors in Table 2.

The models had cross-validated accuracies of 82.2%, 86.1%, and 75.3% for three, four or five bins, respectively. Cohens’s Kappa [113] was 0.731, 0.812, and 0.688, respectively. Due to the slightly greater accuracy, the four-bin model was selected (Figure 2): 0–12 h, >12 h to 1 week, >1 week to 60 days, and >60 days. The four-bin model has an error rate of 11%. The misclassification events for the four-bin model were near the margins of the bins (Figure 2), and only occurred for rat for perfluorooctanoic acid (PFOA) and perfluorononanoic acid (PFNA), and perfluorodecanoic acid (PFDA) in mouse.

Renal elimination includes three processes: glomerular filtration, proximal tubular secretion, and proximal tubular resorption [26]. The mechanistically motivated descriptors initially considered were selected to provide surrogates for PFAS-specific mechanisms of toxicokinetics, with an emphasis on potential renal resorption by the proximal tubules [96]. We do not know the species- and chemical-dependent affinities for the transporters driving section/reabsorption, nor the expression levels of the transporters. We do know that some transporters are expressed along the surface of the proximal tubule. Thus, we can assume that geometry might potentially be correlated with expression level. Similarity to the endogenous ligands of those transporters provides a potential correlate of affinity. The importance of predictors was estimated by the decrease in model performance when the predictor was randomized [109]. The five most important predictors (Table 3) were the average mass of the compound; OPERA model predictions for the logarithmic Octanol:Air partition coefficient and Vapor Pressure; and the kidney descriptors Glomerular Surface Area (SA):Kidney Weight Ratio and Proximal Tubule Diameter. In the case of average molecular mass, a recent review of *t*_½_ data found that PFAS *t*_½_ tends to increase with molecular weight in the same species included in this study [114]. This is consistent with previously observed increases in PFAS *t*_½_ with increasing carbon chain length [26,27,36]. The belief that shorter chains result in faster excretion has prompted a drive to develop alternative chemicals with shorter carbon chains. For example, the chemical GenX is branched and has a shorter *t*_½_ than straight chain PFAS, though without more data we do not know if this generalizes across PFAS.

We found surrogates for active transport among the predictors. First, the kidney physiology predictors are likely proxies for both physical differences and species variation in the expression of transporters for PFAS. The kidney is a primary site of PFAS disposition and elimination for the body [96,99]. Previous work shows that anionic transporters play a key role in renal excretion and reabsorption of PFAS compounds [26]. Renal transporters reside on the membrane of the proximal tubules [26]. Importantly, proximal tubule structural features (length, surface area) were strongly correlated with body weight. Body weight was used to predict proximal tubule structural features for species for which data were not available (monkey and mouse). These results are, therefore, supportive of the need to further understand renal transporter activity for PFAS across species to better extrapolate to humans. Endogenous ligand similarity was the second type of surrogate for active transport that we considered. Three distinct endogenous ligands were identified after the others were eliminated based on correlation to these three. PFAS similarity to hexanoic acid (DTXSID7021607, CAS 142-62-1), butanoic acid (DTXSID8021515, CAS 107-92-6), and heptanedioic acid (DTXSID5021598, CAS 111-16-0) were considered as a surrogate for transporter affinity. Inclusion in our model indicates that the kidney transporters for which these compounds are ligands may be involved in PFAS *t*_½_.

### 3.2. Model Evaluation

The aim of supervised machine learning is to identify patterns of descriptor values that predict how each entry in the training set has been “labelled”. Here, we labelled each measured *t*_½_ according to a broad bin (or category) spanning a range of times. To evaluate whether the patterns occur by chance, we used y-randomization. Additional models were constructed, following the same procedure as above but using ten y-randomized datasets. In a y-randomized dataset all descriptors were held the same, but the bins for the *t*_½_ values were randomly permuted. The predictive performance of the ML model presented was compared to the performance of multiple y-randomized models. The non-randomized ML model accuracy (86.4%) was better than any of the models constructed with y-randomized data. A model using *t*_½_ values randomized across all species-by-PFAS combinations had low predictive value (accuracy of 32.2 ± 13.3%).

y-Randomization showed that some variation in *t*_½_ is accounted for by differences at the species and chemical level. The models for *t*_½_ with training data randomized within species but not chemicals (that is, the chemicals were correct) had an accuracy of 36.8 ± 13.4%. The models where training data chemical identities were randomized, but not species, had an accuracy of 50.2 ± 15.6%. That is, species-specific data alone provide information about the plausible values of *t*_½_ of PFAS. However, the large improvement (86.4% vs. 50.2% accuracy) of the fully non-randomized model suggests that enough chemical-species TK interactions exist to justify combining chemical and species information together. The improvement of the full model over any randomized model indicates that the presented model for *t*_½_ does not occur by chance.

The no information rate is an additional effective “null hypothesis” that we examined. The no information rate is the accuracy for a model that predicts all chemicals to be in the most common bin. The four-bin model has an accuracy of 86.4% compared to the no information rate of 27%. That is, the accuracy of the model presented here is an improvement over selecting the most commonly occurring bin. Since 64% of human *t*_½_ falls into Bin 4 (the longest *t*_½_), this provides a species-specific no information rate. The model accuracy (100% for humans) is greater than the human no information rate. The prevalence of predicted Bin 4 chemicals for humans across other PFAS (56%, as discussed in the following section) indicates fewer long *t*_½_ PFAS than would be expected from the human observations alone.

### 3.3. Application of the Model to a PFAS Library

For each chemical–species prediction, the median half-life of the training data in each bin was used as the predicted *t*_½_. For Bin 1 (<12 h) the median was 4.9 h; for Bin 2 (<1 week) 2.2 days; for Bin 3 (<2 months) 33 days; and for Bin 4 (>2 months) the median used was 3.3 years.

#### 3.3.1. *t*_½_ Predictions for CCD PFAS List

In Figure 3, we show predictions of *t*_½_ across species and sex. Of the 8163 PFAS on the CCD PFAS master list, 6603 had sufficient information for model application. The applicability domain (AD) characterizes the range of chemicals for which we expect accurate predictions [115]. Using the method of Roy, Kar and Ambure [111], we found that the majority (63%) of these chemicals fall into the domain of the model. Across the four species, 4136 PFAS were within the AD (Figure 3A). For humans (over both sexes and dosing methods), 3890 chemicals were estimated to be within AD. Of these, 56% were classified in *t*_½_ Bin 4, 7% were classified in Bin 3, and 37% were classified in Bin 2. We can further restrict predictions to only those chemicals within the ADs of the OPERA models (described as the AM domain; that is, intersection of All Model ADs). The AM domain further reduces the list to 2645 of the 6603 chemicals. For humans, a majority (47%) of this subset of chemicals were predicted to fall into Bin 4, followed by 45% in Bin 2 and 9% in Bin 3. Using the ClassyFire chemical structure ontology [56], the training set could be split into three classes: alkyl halides (9 chemicals), carboxylic acids and derivatives (GenX), and organic and sulfonic acids and derivatives (F-53B). A total of 921 of the PFAS were in these three classes (Figure 3B). For humans, a majority (60%) of this subset of chemicals were predicted to fall into Bin 4, followed by 34% in Bin 2 and 5% in Bin 3.

For chemicals in the domain, *t*_½_ values tended to increase with relative body size. Mice (0.022 kg) and rats (0.225 kg) had more *t*_½_ values in the two fastest bins. Monkeys (3.8 kg), humans (70 kg), and dogs (20 kg) tended to have *t*_½_ values in the three slower bins (Figure 3A). When considering only those chemicals in the AM domain that also align with the three ClassyFire-based classes of the training set (Figure 3B), a similar pattern associated with body size emerges.

The differences in *t*_½_ predictions between species are driven by those parameters in Table 3 that vary between species. From most to least important, these are Glomerular Surface Area (SA) to Kidney Weight Ratio, Proximal Tubule Diameter, and Glomerular Surface Area to Proximal Tubule Volume Ratio. We note that, while overall body weight was included as potential descriptor, it was eliminated during the variable selection process for being highly correlated with these more informative parameters. While these parameters explicitly describe the geometry of the kidney nephron, they are also potential surrogates for multiple aspects of TK. Geometry impacts the flow through the nephron and extent of glomerular filtration, both of which can, in turn, impact the efficiency of clearance of PFAS from the blood. Additionally, since secretion/resorption transporters line the surface of the proximal tubule, the geometry of the proximal tubules (amount of surface area) provides an upper limit on the amount of transporter expression. Albumin binding affinity has the potential to vary between species [116], but species-specific data were not available for enough chemical–species combinations to be used as descriptors here. The number of chemicals within the domain of applicability in Figure 3 varies between species. This is because we calculated the domain of applicability as a function of both chemical and species descriptors, so that similarity to the training set depends on the specific PFAS-species combination.

For humans, no chemicals were predicted to fall within the fastest bin (<12 h). For Perfluoroundecanoic acid (PFUnDA, DTXSID8047553), Zhang, Beesoon, Zhu and Martin [35] observed a half-life of 12 years for men and a half-life of 4.5 years for women. Our model correctly predicted the longest bin (>60 days, median 3.32 years) for both sexes. PFUnDA was not included in our data set because a value was available for humans only.

The model predicts that two ether PFAS, Perfluoro-2,5-dimethyl-3,6-dioxanonanoic acid (DTXSID00892442) and Perfluoro(2-((6-chlorohexyl)oxy)ethanesulfonic acid (DTXSID80892506), are bioaccumulative, but that a third is not (Perfluoro-2-methyl-3-oxahexanoic acid, DTXSID70880215). These predictions for ether PFAS are consistent with fish bioconcentration factors for these three chemicals [117].

#### 3.3.2. Prediction of Whole-Body Clearance and Steady-State Concentration

*t*_½_ predictions were combined with an estimate of *V_d_* to calculate steady-state concentrations (*C_ss_*) using Equations (1)–(3). The application of our ML methodology did not support a model for *V_d_* (see Appendix A). Instead, for all PFAS, we used the median across ~100 PFAS-by-species measurements (see Appendix A), 0.202 L/kg bodyweight. Based on the available kidney descriptors [101], we made predictions for a total of eight species (human, cattle, chicken, dog, horse, monkey, mouse, rabbit and rat) across chemicals falling into the AM domain. Clearance predictions in units of L/kg bodyweight/day are provided by column “CLtot.Lpkgbwpday” in Appendix A. We anticipate that these predictions may be useful in cross-species extrapolation [118]. For humans, predictions for the chemicals PFOS and PFOA fell into same *t*_½_ bin, corresponding to an average clearance of 1.15 × 10^−4^ L/kg BW/day. The 2016 EPA Drinking Water Health Advisories used 8.1 × 10^−5^ and 1.4 × 10^−4^ L/kg BW/day for these chemicals, respectively [42]. Those values were calculated using measured estimates of *t*_½_ from exposed populations and similar values of V_d_ (0.23, 0.17 L/kg bw) [42]. Thus, model predictions for these chemicals fell reasonably close (that is, within an order of magnitude) of values calculated using measured data. PFOS and PFOA are the only two PFAS for which regulatory clearance estimates are available at the time of this analysis.

#### 3.3.3. Domain of Applicability

Based on the range of properties of the training data (using the method of Roy, Kar and Ambure [111]), we found that 4136 PFAS were within the AD. Restricting predictions to only those chemicals whose properties were within the ADs of the OPERA predictors reduced this to 2645 PFAS. Alternatively, using the ClassyFire chemical structure ontology [56] restricted predictions to 921 PFAS. Expansion of the AD will require additional PFAS data both for *t*_½_ and physico-chemical properties. It is hoped that both the *t*_½_ model predictions and estimated AD can guide the selection of candidates’ PFAS for additional testing.

Many PFAS are anions at physiologic pH. The distribution coefficient LogD characterizes the extent to which ionization impacts tissue partitioning. Initial work showed that LogD (as predicted by OPERA), which describes the distribution of substances as a function of lipophilicity and ionization state, was a predictor of *t*_½_ [119]. Unfortunately, most of the PFAS without *t*_½_ were calculated to not fall within the AD as calculated from the training set with respect to LogD. Thus, omitting LogD as a descriptor here slightly reduced the final model accuracy (from 87.2% to 86.4%), but increased the number of chemicals for which predictions could be made (from 1598 to 4136).

Though ionization has often been considered in drug development [120,121], the treatment of ionization equilibria has typically lagged for non-pharmaceutical chemicals [122]. Instead, success has been found considering other aspects of distribution [123,124,125]. The presence of molecular fluorines is thought to increase bioavailability through the modulation of ionization in medicinal chemistry [126]. Unfortunately, neither proprietary nor open-source ionization models include many PFAS in their training sets because the data to do so do not yet exist. Additional measurements for PFAS with more varied LogD might enhance predictivity and provide an evaluation of whether this is an actual applicability domain issue. Other, similar issues are expected to be identified as the *t*_½_ data are expanded. Ultimately, environmental decision makers may not have the luxury of waiting for more data, but might rather identify suitable chemical analogs [127]. It is hoped that this model provides a tentative tool for classifying PFAS TK *t*_½_ on the basis of four bins of “analog” PFAS.

Using the ClassyFire chemical structure ontology, the total set of 6603 PFAS spanned 150 classes. As we calculated AD based on predictor values, the subset within the *t*_½_ model domain spanned 149 classes, and the further subset within the AM domain spanned 121 classes. Alkyl halides made up the largest class of both subsets, with (14%). The other two training set classes, carboxylic acids and derivatives, and organic sulfonic acids and derivatives, made up 9% and 3%, respectively. As estimated from the predictors, there are diverse PFAS included in the *t*_½_ model and AM domains, despite the narrow training diversity employed. This suggests that the predictors included were successful in capturing key drivers of *t*_½_ variability. The most commonly occurring classes that were within the domain of the predictor values but that were not represented in the training set were organofluorides (13%), organooxygen compounds (11%) and fatty acyls (7.5%). These classes make good targets for future data collections. PFAS chemicals outside the AD included 44 classes, with the largest class (17% of chemicals) consisting of benzene and substituted derivatives.

See Appendix A for model predictions, applicability domain status, ClassyFire classifications, and steady-state TK predictions for all CCD PFAS list chemicals for which sufficient information was available for model application. All the code to reproduce models and results is available from: https://github.com/USEPA/CompTox-PFASHalfLife (accessed 17 January 2023).

### 3.4. Model Limitations and Future Considerations

The knowledge of PFAS TK is essential for risk assessment of this large and important class of chemicals. Chemicals with longer *t*_½_ may bioaccumulate, and thus may warrant closer regulatory scrutiny. The majority (56%) of PFAS were predicted to be in the longest *t*_½_ category in humans. This study is an initial attempt to use ML to organize existing data to inform the TK of unmeasured PFAS. The accuracy (86.4%) of the ML developed here was far better than expected by chance (y-randomized accuracy was 32.2 ± 13.3%). While the constructed model was successful in describing the large variation in *t*_½_ values of the training set (Figure 2) across species and chemicals, its development made use of most of the data available in the published literature. The training set consisted of only four species and 11 chemicals, and was dominated by alkyl halides; namely, perfluoro-carboxylic acids and perfluoro-sulfonic acids. The chemical structural space of the predicted chemicals within the AM domain was much more diverse than the training set. The distribution of *t*_½_ was more heavily weighted toward faster values when chemicals were subset to contain only the three classes of chemicals found in the training set (Figure 3B). If the TK behavior of other classes of PFAS are significantly influenced by factors not captured by the included predictors, then predictions could be unreliable for those chemicals. These uncertainties can only be disentangled with additional data to evaluate this or similar models.

*t*_½_ alone is insufficient to predict the TK of PFAS, including peak and time-integrated plasma concentrations (respectively, *C_max_* and area under curve/AUC). Even the simplest approaches to TK modeling (that is, the one compartment empirical model) require the parameter *V_d_*. Despite compiling a dataset of ~100 PFAS-by-species measurements of *V_d_*, our ML model-building approach was unsuccessful (see Appendix A). In comparison to *t*_½_, the compiled values for *V_d_* varied relatively little. Median *V_d_* values ranged across chemicals from 0.139 to 0.368 L/kg, and across species from 0.194 to 0.254 L/kg. Thus, our failure to build more compelling models for predicting inter-chemical and -species differences in *V_d_* is at least partially a function of the lack of variability among the data relative to the strong uncertainty. Notably, the uncertainty in the literature measurements just for PFOS in rat included *V_d_* that ranged from 0.09 to 7.0 L/kg. This broad uncertainty confounded attempts to build a ML model. However, it possible to use the species-specific predictions provided in Appendix A to make TK predictions for PFAS (including *C_max_* and AUC) using the median dataset value of *V_d_* (0.201 L/kg), as we did in Section 3.3.2. In addition, some PFAS have the potential to transform in vivo to a variety of metabolites—which often include the 11 PFAS modeled here [128]. Thus, the development of data and models to predict metabolites that could be coupled with models of *t*_½_ may greatly enhance our ability to predict the TK of PFAS.

Physico-chemical properties and albumin binding can both be measured. While OPERA has recently incorporated measurements for PFAS into its QSARs, additional measurements of PFAS albumin binding may be extremely useful both on their own and as training data for QSARs. Similarly, the critical micellular concentration (CMC) could be measured. CMC is a property that characterizes the aggregation of a chemical into micelles of like molecules, a process that essentially sequesters the chemical away from the rest of the body. For PFAS, the formation of a fluorous-phase of micelles is potentially irreversible [129], and might result in longer *t*_½_ with decreasing CMC. Observed PFAS CMC tend to exceed 10 mM [130] and it remains unknown whether fluctuations or gradients ever lead to such concentrations physiologically. Regardless, experimental values for the CMC are not widely available and the only predictor for CMC available at the time of this writing was omitted for being based on proprietary descriptors [130]. Therefore, the creation of an open-source, verifiable model for CMC could provide an additional relevant PFAS descriptor for predicting TK.

Ample opportunity remains in both the experimental and epidemiological domains for researchers to generate the data with which to test these model predictions, as well as develop alternative models and descriptors. Figure 3 suggests that future in vivo TK studies in rodents might aim to investigate PFAS that are predicted to have different half-lives. This will allow the evaluation and refinement of ML approaches such as those developed here, as well as informing TK study design (for example, if measuring concentration changes over weeks is required). Taken as a synthesis of the available data on PFAS TK data, the prediction of our model might also help analysis of future TK studies by providing informative Bayesian priors [14,16,131]. Similarly, human studies investigating exposure changes (such as switching water supplies) might target unmeasured PFAS predicted to clear more rapidly. Finally, analysis of human biomonitoring data might qualitatively look for greater accumulation (that is, higher observed concentrations) of PFAS predicted to have long *t*_½_ as opposed to those PFAS predicted to clear rapidly in humans.

We hope that the resources presented here will be used as a starting point by the broader scientific community to develop additional data and models for PFAS TK. The term “forever chemical” has been applied to some PFAS with regard to their persistence in the environment, bioaccumulation, and long human half-lives [132]. For humans, this preliminary model distinguishes between those PFAS with *t*_½_ greater than two months and those that are eliminated much faster from the body. “Forever” lurks among those longer *t*_½_ PFAS.

## Figures and Tables

**Figure 1 toxics-11-00098-f001:**
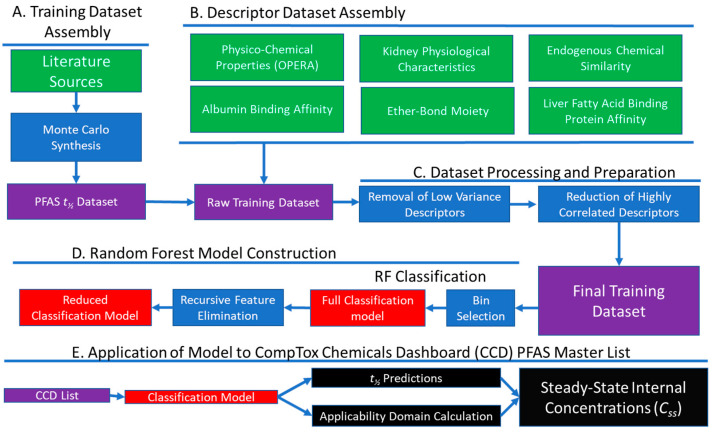
Scientific workflow including (**A**) Training Data Assembly, (**B**) Predictor Dataset Assembly, (**C**) Dataset Processing and Preparation, (**D**) Random Forest Model Construction, and (**E**) Application of the Models to the CCD PFAS list. Green boxes denote data sources, purple boxes denote assembled datasets, red boxes denote models, blue boxes data denote processing steps, black boxes denote model outputs, and arrows indicate flow between steps.

**Figure 2 toxics-11-00098-f002:**
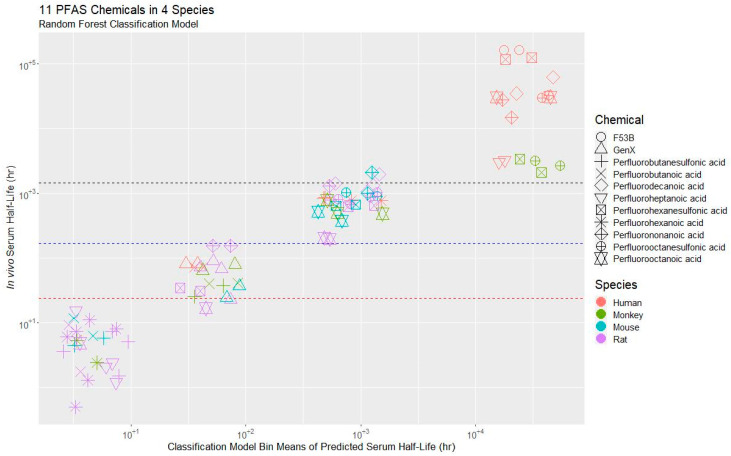
Values of *t*_½_ of the training data (*y*-axis) vs. classification predictions by the RF Classification model using 15 predictors. Colors signify species, while shapes indicate different PFAS compounds. Bin margins (<12 h, 12 h^−1^ week, 1 week^–2^ months, >2 months) are indicated as dotted lines. Note that observations have been jittered (that is, a small amount of random variation has been added) along the *x*-axis to increase readability.

**Figure 3 toxics-11-00098-f003:**
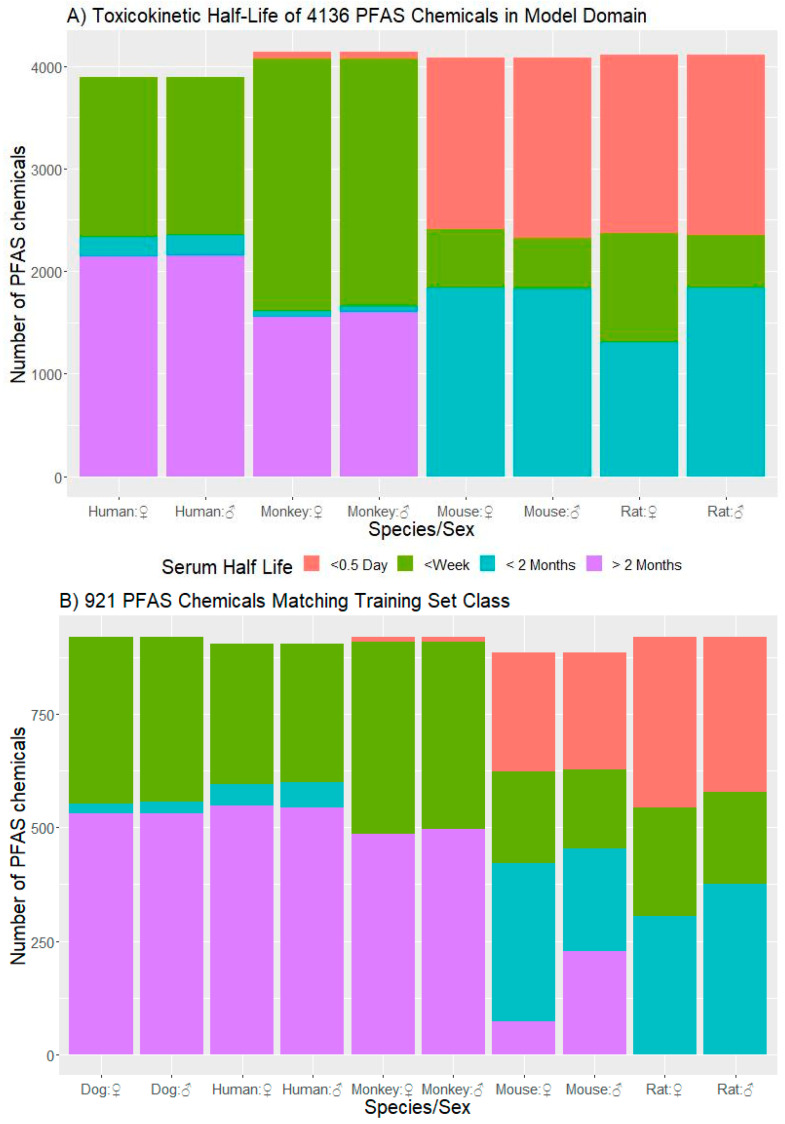
Distributions of predicted *t*_½_ for (**A**) 4136 PFAS within the AD of the RF Classification model, and (**B**) 921 PFAS classified in the same 3 classes as the 11 training set chemicals via ClassyFire. Shown are the number of chemicals predicted to fall within half-life categories by sex (male = ♂, female = ♀) for 5 species. Bins are denoted by color, with pink ≤ 12 h, green = 12 h^−1^ week, blue = 1 week^−2^ months, and purple ≥ 2 months.

**Table 1 toxics-11-00098-t001:** PFAS *t*_½_ life estimates used in model construction (full data set is provided in Appendix A). Data adopted from Fenton, Ducatman, Boobis, DeWitt, Lau, Ng, Smith and Roberts [11] was augmented by new studies wherever available. Values for chemical/species combinations that were not available were omitted from modeling, but values only available for one sex of a species were assumed to be same for both.

		Rat			Mouse			Monkey			Human		
		(*Rattus rattus*)			(*Mus musculus*)			(*Macaca fascicularis*)			(*Homo sapiens*)		
Chemical CAS/DTXSID	Sex	Value	Unit	Ref.	Value	Unit	Ref.	Value	Unit	Ref.	Value	Unit	Ref.
PFBS (C4) 375-73-5 DTXSID5030030	F	1.5–7.4	Hours	[34,73,74]	4.5	Hours	[75]	1.1	Days	[73,74]	35	Days	[36,73]
	M	3.6–5.0			5.8			1.6			36		
PFHxS (C6) 355-46-4 DTXSID7040150	F	1.3–1.4	Days	[34,76,77]	27	Days	[76]	87	Days	[76]	13	Years	[35,36,37,39]
	M	26–27			28			140			14		
PFOS (C8) 1763-23-1 DTXSID3031864	F	28–43	Days	[32,34,77]	38	Days	[32]	110	Days	[32]	3.4	Years	[35,36,37,38,39]
	M	34–36			43			130			3.7		
PFBA (C4) 375-22-4 DTXSID4059916	F	1.8	Hours	[78]	6.2	Hours	[78]	1.7	Days	[78]	3	Days	[78]
	M	9.2			12								
PFHxA (C6) 307-24-4 DTXSID3031862	F	0.5–7.3	Hours	[74,79,80,81]				2.4	Hours	[74]	32	Days	[31]
	M	1.3–11						5.3					
PFHpA (C7) 375-85-9 DTXSID1037303	F	1.2–2.1	Hours	[25,79]							140	Days	[35,36]
	M	1.5–2.4									130		
PFOA (C8) 335-67-1 DTXSID8031865	F	1.7–4.8	Hours	[25,77,80,82]	16	Days	[83]	33	Days	[84]	3.5	Years	[35,36,37,85]
	M	8.1–8.5	Days		22			20–21					
PFNA (C9) 375-95-1 DTXSID8031863	F	6.4	Days	[25,86,87]	42	Days	[87]				1.7	Years	[35]
	M	3.3–5.5			87						3.2		
PFDA (C10) 335-76-2 DTXSID3031860	F	45–59	Days	[25,80,86]							4	Years	[35]
	M	55–83									7.1		
F-53B 756426-58-1 DTXSID80892506	F										18	Years	[88]
	M												
GenX 13252-13-6 DTXSID70880215	F	0.9–2.8	Days	[89]	1.0	Days	[89]	3.3	Days	[89]	3.4	Days	[90]
	M	3.0–3.7			1.5			2.7					

**Table 2 toxics-11-00098-t002:** Summary of Descriptor Set Used. (A) All chemical descriptors used in model construction. * = indicates value is the mean, rather than median. This was used for binary descriptors with either a 1 or 0. For ether bond: 1 = present, 0 = not present; for endogenous similarity measures, 1 = similar (≥90% Tanimoto score), 0 = not similar. Endogenous ligand similarity was included as a surrogate for chemical-specific transporter data. (B) All physiological descriptors included in model by species. As additional surrogates for kidney transporter data, we focused on the geometry of the proximal tubule where they are expressed. Body and kidney weight (italicized) included here for reference but were identified as highly correlated with other features and eliminated by feature reduction for model building. (C) Categorical descriptors used.

**A–Chemical Structure Descriptors**					
**Parameter Type**	**Descriptor**	**Chemical Coverage (%)**	**Training Set Median**	**Training Set Min**	**Training Set Max**
Protein binding	Albumin binding affinity constant (Mol^−1^)	45.45	2.84 × 10^5^	2800	1.10 × 10^6^
Physico-chemical	Average Mass (g/mol)	100	400.1	214	532
	Log Vapor Pressure (mmHg)		−2.07	−8.09	1.53
	Log Octanol: Air		4.16	3.46	6.33
	Log Octanol: Water		3.11	1.43	5.61
	Log Water Solubility (Mol/L at 25 °C)		−2.68	−4.9	−0.5
	Ether bond present		0.13 *	0	1
Endogenous Ligand Similarity	CAS 142-62-1	100	0.18 *	0	1
	CAS 107-92-6		0.088 *		
	CAS 111-16-0		0.066 *		
**B–Physiological Descriptors**					
**Species**	**Proximal tubule diameter (mm)**	* **Body Weight (kg)** *	* **Kidney Weight/Body Weight (g/kg)** *	**Glomerular Surface Area/Proximal Tubule Volume (1/mm)**	**Glomerular Surface Area/Kidney Weight (mm^2^/kg)**
Human	0.072	*70*	*2.23*	3.16	1.65
Monkey	0.062	*5*	*2.5*	2.13	2.04
Mouse	0.054	*0.02*	*8*	2.05	2.28
Rat	0.058	*0.24*	*2.92*	2.31	3.26
**C–Categorical Descriptors**					
**Sex**	Female/Male				
**Dosing**	intravenous, oral, other (epidemiological, via metabolite extrapolation)				

**Table 3 toxics-11-00098-t003:** Predictor Importance [109] (percent reduction in accuracy) of all model predictors.

Parameter	Raw Accuracy Change	Scaled Accuracy Change
Average mass	9.49	100
Log Octanol:Air (OPERA)	7.02	73.3
Glomerular Surface Area (SA): Kidney Weight Ratio	6.32	65.6
Proximal Tubule Diameter	6.11	63.4
Log Vapor Pressure (OPERA)	4.86	49.7
Log Octanol:Water (OPERA)	4.37	44.4
Glomerular Surface Area: Proximal Tubule Volume Ratio	4.14	42.0
Log Water Solubility (OPERA)	3.72	37.4
Dosing Form	3.26	32.4
Albumin binding affinity	3.16	31.3
Ether Bond (COC)	2.56	24.8
Sex	2.14	20.2
Similarity to CAS 142-62-1	1.93	18.0
Similarity to CAS 107-92-6	0.61	3.63
Similarity to CAS 111-16-0	0.27	0

## Data Availability

An R-markdown file allowing the application of the model to novel chemicals and species is available for download at the following git repository: https://github.com/USEPA/CompTox-PFASHalfLife (accessed 17 January 2023).

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
