# Peer review of "A Machine Learning Model to Estimate Toxicokinetic Half-Lives of Per- and Polyfluoro-Alkyl Substances (PFAS) in Multiple Species"

_toxics, 2023, doi:10.3390/toxics11020098_

Round 1

Reviewer 1 Report

To predict t½ values and a simple TK model to predict whole body clearance and steady-state plasma concentrations in multiple species, the authors used the random forest method to develop a machine learning model for PFAS t½. They classified PFAS chemical/species t½ into four categories: less than 12 hours, 12 hours to 1 week, 1 week to 2 months, or greater than 2 114 months.

Comments

-line 129, The described webpage  “https://github.com/USEPA/CompTox/PFASHalfLife is not usable, please check again.

-Table 2, Glomerular Surface Area / Proximal Tubule Volume and Glomerular Surface Area / Kidney Weight are dimensionless, please give dimension. If they are dimensionless, please explain why

Although authors give their validity of the machine learning to predict TK models, do you have any method to validate your machine learning TK model?

Reviewer 2 Report

The authors described a machine learning method based on random forest to model the existing t½ across 4 species and 11 PFAS and predicted t½ of 3890 compounds within domain of the model. This work is interesting, however, the manuscript still needs a major revision before reconsideration for publication in Toxics.

Major points:

1. Whether the sample size meets the training requirements of the random forest method?
2. Why the PFAS t½ data fit ML-based approach? This part was logistically confused and needs to be re-elucidated in the Introduction.

3. Pharmacokinetic parameters, such as AUC, Cmax, may be more important potential predictors than other descriptive variables, but the manuscript did not mentioned this point.

4. How the animal species causes the difference in distributions of predicted t1/2? It should be discussed in the Results.

5. The information of R packages should be written in the Materials and Methods.

Minor points:
1.In line 76, there was an incorrect line break.

Round 2

Reviewer 2 Report

The authors addressed the concerns well and I recommend to accept this manuscript.